# Neuroimmune Response in Natural Preclinical Scrapie after Dexamethasone Treatment

**DOI:** 10.3390/ijms21165779

**Published:** 2020-08-12

**Authors:** Isabel M. Guijarro, Moisés Garcés, Belén Marín, Alicia Otero, Tomás Barrio, Juan J. Badiola, Marta Monzón

**Affiliations:** Research Centre for Encephalopathies and Transmissible Emerging Diseases - Institute for Health Research Aragón (IIS), University of Zaragoza, C/Miguel Servet 155, 50013 Zaragoza, Spain; isabelmariagt91@gmail.com (I.M.G.); moisesgarces1@gmail.com (M.G.); belenm@unizar.es (B.M.); aliotgar@hotmail.com (A.O.); tbarrio_sc@hotmail.com (T.B.); badiola@unizar.es (J.J.B.)

**Keywords:** scrapie, dexamethasone, glucocorticoids, cytokine, neuroinflammation, neuroglia, preclinical stage, prion diseases

## Abstract

A recently published report on chronic dexamethasone treatment for natural scrapie supported the hypothesis of the potential failure of astroglia in the advanced stage of disease. Herein, we aimed to extend the aforementioned study on the effect of this anti-inflammatory therapy to the initial phase of scrapie, with the aim of elucidating the natural neuroinflammatory process occurring in this neurodegenerative disorder. The administration of this glucocorticoid resulted in an outstanding reduction in vacuolation and aberrant protein deposition (nearly null), and an increase in glial activation. Furthermore, evident suppression of IL-1R and IL-6 and the exacerbation of IL-1α, IL-2R, IL-10R and IFNγR were also demonstrated. Consequently, the early stage of the disease is characterized by an intact neuroglial response similar to that of healthy individuals attempting to re-establish homeostasis. A complex network of neuroinflammatory markers is involved from the very early stages of this prion disease, which probably becomes impaired in the more advanced stages. The in vivo animal model used herein provides essential observations on the pathogenesis of natural scrapie, as well as the possibility of establishing neuroglia as potential target cells for anti-inflammatory therapy.

## 1. Introduction

Scrapie is the prototype of prion diseases, and is one of the most common neurological diseases in sheep, together with coenurosis and leptospirosis, among others [1].

Several reasons prompted us to perform this study on the neuroinflammatory response of naturally affected animals in the preclinical stage of scrapie. Firstly, neuroinflammation is an intrinsic concept in the neurodegenerative process [2,3], in the context of which cytokines have gained huge attention because they have been demonstrated to be involved in the inflammatory processes leading to neurodegeneration [3,4,5]. Secondly, this complex immune reaction is carried out by neuroglia in this and other neurodegenerative processes. Nevertheless, whether it really plays a neurotoxic [6,7,8,9] or neuroprotective [10,11] role is still unknown. Thirdly, this host response has been demonstrated to be specifically involved in the neurodegeneration of prion diseases in sheep [12,13,14], coinciding with the onset of clinical signs in the murine model [15] and in Creutzfeldt Jakob disease (CJD) [6]. Cytokines have been proven to be relevant mediators in cellular communication in several models of this group of diseases [6,7,8,12,15,16,17].

Within this framework, we recently characterized the glial activation and response in clinical ovine scrapie after chronic anti-inflammatory treatment with dexamethasone (DEX) [14], demonstrating the potential failure of astroglia at that stage of scrapie. One of the advantages of this in vivo model is the possibility of extrapolating the conclusions drawn from this to other prion [18,19] and prion-like diseases [20] in a feasible manner. Studies in murine models are sometimes contradictory, and they have not been able to reach a consensus regarding inflammatory responses [6,7,15,21]. 

Our previous results regarding the glial response and survival period in the clinical stage prompted us to study earlier stages of the disease [14]. Additionally, a previous study described the presence of several genes involved in inflammation that are up-regulated in the early phases of prion infection in mice [22]. Thus, the specific objective of this study was to assess the effects of this same anti-inflammatory treatment, DEX, in natural scrapie, but in the initial phases of the disease. By means of descriptions of neuropathological lesions and glial alterations, both in themselves and in their communication via the cytokines released, a more complete neuroinflammatory profile regarding the progress of natural scrapie was sought. The overall aim was to confirm or discard neuroglia as a potential therapeutic target.

## 2. Results

A summary of the statistical results is shown in Table 1. Because of the high number of markers used in four different brain areas from two different groups, only one case showing evident statistical differences for each marker was selected to illustrate the results. 

As mentioned, no preclinical sheep, regardless of treatment, presented any of the key clinical signs of scrapie throughout the time of the experiment in any of the cases.

### 2.1. Histopathological Findings (H-E)

#### Vacuolation

Vacuolation was mainly located in the neuropil in all cases when it was found, although intraneuronal vacuolation was observed in some of them too. As expected, in regard to differences between brain regions, subtle spongiform changes were found in the frontal cortex (Fc) and cerebellum (Cb), while it was most pronounced and severe in the obex (O) (Figure 1). Strikingly, a statistical decrease in spongiform change was found to be associated with DEX treatment in the O (* *p* = 0.019) (Figure 2A). 

### 2.2. Immunohistochemical Findings (Ihc)

#### 2.2.1. PrP^sc^ Deposits

In general, prion protein accumulation was always observed in lymphoid tissues, but not in nervous tissue. Few pathological prion protein (PrP^sc^) deposits were observed in the brain tissues of untreated animals, and were exclusively in the Cb of one treated preclinical animal. Similar to the histopathological results, a statistical decrease in prion accumulation was also evidenced as a result of treatment in the O and medulla oblongata (MO) (** *p* = 0.004 in both; Figure 1 and Figure 2B). 

#### 2.2.2. Astrogliois

Regarding astrogliosis, all samples from all brain areas coming from DEX-treated sheep showed an evident increase in Glial Fibrillary Acidic Protein (GFAP) immunoreactivity compared with the respective untreated group (strong astrogliosis; Figure 1). The highest increases in astrogliosis between DEX-treated and untreated sheep were observed in the O and MO, showing very significant statistical differences (** *p* = 0.008 and ** *p* = 0.004, respectively; Figure 2C).

Morphologically, GFAP immunolabeling showed a hypertrophic morphology in 100% of treated sheep compared to the astrocytes in untreated animals in all brain regions, which appeared to be more stellate (Figure 3). 

#### 2.2.3. Microgliosis

With respect to microgliosis, IHC for the IBA-1 marker always demonstrated an expansion of the microglial population in preclinical DEX-treated sheep (strong microgliosis) when compared with non-treated animals (Figure 1). In fact, except for the Cb (*p* = 0.08), all brain regions showed a statistically significant increase (* *p* = 0.028, * *p* = 0.035 and * *p* = 0.025) in this glial expansion in DEX-treated sheep (for the O, MO and Fc, respectively) (Figure 2D). 

Regarding microglial morphological changes, a great number of microglial cells demonstrated an amoeboid phenotype after DEX treatment in all brain areas, while a ramified phenotype was the most frequent in the non-treated specimens (Figure 3).

#### 2.2.4. Cytokine Detection

In relation to the neuroinflammatory markers assessed here, our immunohistochemical results revealed that preclinical sheep showed a very significant increase in IL-1 expression in the Fc when they were treated with DEX (** *p* = 0.008; Figure 2E and Figure 4), while no differences were observed in the rest of the brain areas analyzed. Meanwhile, immunostaining for IL-1R was lower in treated animals as compared to the untreated ones, showing a statistically significant difference in the MO (* *p* = 0.05; Figure 2F and Figure 4). Furthermore, the expression of IL-2R increased in all brain areas of preclinical sheep after treatment, and a statistically significant difference was only found in the O (* *p* = 0.05; Figure 2G and Figure 4). On the other hand, immunostaining for IL-6 in DEX-treated sheep was reduced in all brain areas compared to untreated preclinical sheep, and this decrease was strikingly evident in the O and MO (*** *p* = 0.001 in both areas; Figure 2H and Figure 4). Regarding IL-10R, the Mann–Whitney U test revealed the significant effects of DEX in preclinical treated sheep compared to untreated ones, showing an increase in this receptor’s expression in all brain areas. They were significant in the Fc (** *p* = 0.01) and MO (* *p* = 0.02), and very significant in the Cb (** *p* = 0.003), while a trend was observed in the O (# *p* = 0.07; Figure 2I and Figure 4). In addition, immunostaining intensity for TNFαR was the least intense in all brain areas compared to the rest of the neuroinflammatory markers assessed. Moreover, no statistically significant changes in intensity levels were detected for this marker in any brain area after DEX treatment (Figure 2J), it being the only one that did not change its levels with DEX treatment in this sheep model. Finally, DEX-treated sheep displayed a higher intensity in IFNγR immunostaining compared to the untreated ones in all brain areas (Cb, O and MO; ** *p* = 0.006 in all cases), except for the Fc (Figure 2K and Figure 4).

Thus, chronic glucocorticoid treatment applied in sheep in the preclinical stage of scrapie resulted in a remarkable reduction in vacuolation and pathological protein deposition (nearly null), as well as an increase in both astroglial and microglial activation. Additionally, an associated suppression of IL-1R and IL-6, along with an exacerbation of IL-1α, IL-2R, IL-10R and IFNγR, were observed in different brain regions in sheep after treatment. 

## 3. Discussion

The findings presented in our recently published study support an impaired astroglial response in the clinical stage of scrapie, suggesting astroglial paralysis [14]. This is in accordance with what was recently described in the late stages of Alzheimer’s disease (AD) [23]. A possible impairment of communication between microglia and astroglia at this stage of the disease is also suggested [14], emphasizing the relevance of the complex network of neuromodulator peptides in prion diseases.

Neuropathological and immunohistochemical analyses of neuroinflammatory activity in different brain areas from preclinical sheep naturally infected with scrapie were compared with DEX-treated animals in the present study. To address this concern, histopathological findings, pathological prion protein accumulation, glial cell activation and several neuroinflammatory markers were investigated. Consistent with the statement that neuroinflammation in prion diseases is region-dependent [24], it was considered crucial to analyze different encephalic areas: the frontal cortex, cerebellum, obex and medulla oblongata.

This work is part of a larger study aiming to elucidate the neuroinflammatory process involved in the neurodegenerative progress of prion diseases. Here, the previous study regarding the efficacy of anti-inflammatory therapy in scrapie was extended to the early stages of the disease in order to determine its potential effect in relation to the evolution of the neuroinflammatory process in the progress of scrapie. This is based on the fact that the mechanisms underlying cytokine release could be a crucial target for therapeutic approaches in the central nervous system (CNS) in prion diseases [25]. Several other studies examined very early time points in scrapie, but most of them used experimental models in rodents [26,27,28] or in sheep [29], with just a few of them having used the natural model of the disease [30,31,32]. To study natural field cases of scrapie is, in our opinion, essential, because they represent a more feasible source of knowledge than experimental models. Moreover, naturally infected sheep have been demonstrated to constitute a suitable model, not only for prion diseases [33], but also for research into other neurodegenerative disorders [19,20].

In the present study, the main neuropathological lesions associated with the disease were investigated, and we demonstrated the efficacy of DEX in the reduction of vacuolation and particularly PrP^sc^ deposition, which becomes practically null in all areas for all treated animals. Our previous results in the clinical stage of scrapie showed that this treatment was not effective in reducing both vacuolation and prion deposition when neuronal degeneration was advanced [14]. This fact was also reported in clinical cases of AD treated with non-steroidal anti-inflammatory drugs (NSAIDs) [34], which were ineffective because neuronal degeneration had already started. As a consequence, we could speculate that the preclinical stage of scrapie constitutes the therapeutic window in which some treatments would be effective in stopping or at least slowing down the progress of neurodegeneration.

The astrocytic marker used in this study demonstrated a remarkable increase in all brain areas for all animals after anti-inflammatory treatment. Meanwhile, this same treatment showed a down-regulation of astrogliosis when it was applied in the clinical stage of the disease [14]. These observations would reinforce astroglial paralysis in the advanced stages of scrapie, as recently suggested in the late stages of AD [23], while astroglial functions are intact in the earlier stages, similar to healthy subjects.

A relevant astrocytic morphological finding, consisting of the presence of hypertrophic astrocytes in treated animals compared to the stellate morphology in non-treated ones, is described here. This is in total agreement with the morphological change undergone by the control animals in our recent study [14]. This again supports the fact that astroglial functions remain similar in the preclinical and healthy stages, reacting against any stimulus (treatment in this case). Although the study of changes in astroglial morphology is a new field and a very small amount of information is available [35], this hypertrophic appearance is, to our knowledge, here described for the first time in preclinical scrapie. Further studies are required in order to determine the real meaning of this and other shape alterations in the progress of neurodegeneration. 

In this in vivo study, an expansion of the microglial population in DEX-treated animals was also proven, showing an increase in the number of microglia as well as an activated and phagocytic amoeboid phenotype. Other authors previously described similar findings [21,36,37] in association with the expression of neuroinflammatory genes [38]. This initial microglial activation might reflect an attempt to start a protective immune response, hence it already being active and expressing more neuroinflammatory genes in the early phases of scrapie infection. With the progress of the disease, the interglial communication could fail, or even a more robust microglial reaction could exacerbate the neurodegenerative process. 

Thus, the highest levels of intensity of both microgliosis and astrogliosis were observed in the obex and the medulla oblongata. Therefore, a protector neuroinflammatory effect is attributed to anti-inflammatory therapy through the activation of both glial populations in the preclinical phase of the disease. Nevertheless, this drug does not produce the effect in the advanced stage (clinical animals), probably due to the impaired response [14].

Overall, variations in all but one of the neuroinflammatory markers assessed in this study were demonstrated in the preclinical stage after treatment.

Preclinical animals demonstrated an increase in IL-1 expression when they were treated. Given that the early overexpression of IL-1 has been related to a fair extent to astrogliosis [17,39], this fact suggests that the secretion of IL-1 by this glial population is properly carried out even in the preclinical stage, when communication with the microglia is not yet impaired. Moreover, higher levels of this same cytokine were also observed in preclinical studies of AD [40,41,42], thus reflecting similarities between prion and prion-like diseases in aspects of the neuroinflammatory process. 

Contrarily, immunostaining for IL-1R was lower in all areas for animals after treatment. The loss of IL-1R1 gene expression has been shown to increase the survival time of mice during infection with scrapie [43]. Consequently, a beneficial effect of DEX is reaffirmed here in the preclinical stage of scrapie. Nevertheless, it is worth mentioning that the self-renewal of microglia is mediated by this peptide [44]. Thus, DEX could be helping to slow down the progress of the disease in earlier stages, while contributing to the poor communication between the microglia and astroglia populations via the prevention of this process of glial renewal.

It is well known that IL-2R is involved in neuronal development [45,46]. Provided that an increase in this mediator was found in all brain areas of preclinical sheep after treatment, a neuroprotective function for DEX would be proposed here. This could be in terms of promoting neuronal development, especially in the obex, the region most affected by vacuolation and prion deposition, and the brain area where the highest increase in this receptor was demonstrated. There seems to be an attempt to recover the early neuronal loss in the preclinical stage, but this probably fails at some point during disease progression into the clinical and irreversible stages.

A considerable body of data has been compiled regarding IL-6, in which IL-6 is described as a pro-inflammatory cytokine in chronic inflammation models [47] that produces neurologic diseases in mice [48]. Its blocking mechanism has even been proposed as a potential treatment for chronic inflammatory diseases [49]. Indeed, after anti-inflammatory treatment, immunostaining for IL-6 was shown to be reduced in all brain areas compared to untreated preclinical sheep. This reduction was especially evident in the obex and medulla oblongata. Both of these areas are, as justified above, where the highest increases in glial activation and prion protein absence were simultaneously observed, supporting the hypothesis that neuroprotection in these areas was especially notable. Consequently, it can be speculated that the anti-inflammatory drug is neuroprotective at this stage. Moreover, the crucial role of this cytokine in prion [12,16,17] and prion-like disease pathogenesis [50,51] is supported by the results provided here.

The DEX treatment applied here was shown to lead to an outstanding increase in IL-10R in all brain areas of preclinical sheep, which reflects the anti-inflammatory properties of this drug [52]. As a protective role has been attributed to this mediator in prion diseases [53,54], it can be hypothesized that anti-inflammatories represent a therapeutic target in the preclinical stage, though not when clinical signs have already appeared.

On the other hand, DEX treatment produced a huge effect on IFNγR expression in preclinical sheep. Taking into account that IFNγ stimulates microgliosis and microglial division [38], this again suggests a neuroprotector role for DEX. Nevertheless, it has also been described as leading to neuronal and glial cell damage [55], possibly underlying the mechanisms of neurodegeneration in more advanced stages of the disease. This hypothesis concurs with the initial activation of microglia observed in the preclinical stage, reflecting an attempt at a protective response but exacerbating neurodegeneration in the clinical stage, possibly mediated by this cytokine.

No changes in levels of TNFαR were detected in any brain area after DEX treatment in preclinical sheep. Furthermore, its immunostaining pattern was the least intense in all brain regions compared to the rest of the neuroinflammatory markers assessed in this study. Consequently, we discarded the possibility that this mediator has a specific role in scrapie progress, as other authors suggested [56].

Overall, in the present study, the glucocorticoid administered resulted in a clear suppression of IL-1R and IL-6. Chronic exposure to this drug has been demonstrated to result in the suppression of inflammatory cytokines; however, it is worth mentioning that this therapy can also potentiate such immunity [57], especially in the CNS [58]. Thus, herein, an exacerbation of the intensity levels of IL-1, IL-2R, IL-10R and IFNγR was observed after chronic exposure to DEX, as was described in murine cases of AD treated with DEX [59]. Actually, immune responses in prion diseases have been described to differ from typical innate immune responses, given that fewer cytokines are elevated and their levels are lower than in conventional CNS infections [60,61]. Therefore, on the basis of the results provided herein, it is suggested that the early scrapie infection is characterized by an intact neuroglial response that tries to re-establish homeostasis, although this probably becomes impaired at more advanced stages of the disease.

The in vivo animal model used here provided fundamental observations on the pathogenesis of natural preclinical scrapie, confirming that a complex network of neuroinflammatory markers is involved in the neurodegenerative process from the very early stages of infection. The results presented here show that the cytokine network in the encephalon of scrapie-affected sheep varies with the progress of the disease, and the complexity of this process needs further study for clarification.

## 4. Materials and Methods 

### 4.1. Animals

The Ethical Committee of University of Zaragoza approved the following experimental procedures (Comisión Ética Asesora para la experimentación animal, 28 Sep 2016. REF: PI41/16). We attempted to minimize animal suffering during the experiments and reduce the number of animals used in all cases.

A total of 12 female Rasa *Aragonesa* crosses pre-clinically affected with scrapie were included in the present study. Two different groups were studied: non-treated (*n* = 5) and DEX-treated (*n* = 7) animals. A summary of cases with details of age, genotype for the PRNP gene, and treatment are shown in Table 2.

These preclinical animals came from monitored flocks belonging to the Spanish Scrapie Surveillance Program, and their status was confirmed by the presence of pathological prion protein in recto-anal mucosa associated lymphoid tissue (RAMALT) biopsies (Figure 5A,B). In a few cases, it was necessary to perform Protein Misfolding Cyclic Amplification (PMCA) and Dot Blot (DB) techniques on lymphoid tissues in order to confirm the in vivo diagnosis (Figure 5C). These animals did not exhibit any clinical signs of scrapie before inclusion in the study or throughout the experiment. An exhaustive clinical diagnosis based on classical signs such as pruritus, tremor, locomotor incoordination and behavioral changes was carried out on all of them.

Treated preclinical sheep were intramuscularly injected daily with DEX (SYVA, León, Spain; 0.04 mg/kg) in alternating posterior limbs after a one-week period of acclimation and until euthanasia by human endpoint criteria due to the side effects of DEX (7 months in the shortest and 17 months in the longest case). Additionally, a daily dose of 0.5 mg/kg of omeprazole was administered to all sheep in order to avoid the appearance of gastric ulcers. 

Necropsy of each sheep was performed after euthanasia with pentobarbital injection. A total of 80 samples were collected (including different tissues: digestive, lymphoid, nervous and other viscera) and each one was fixed by immersion in 4% paraformaldehyde for this and other distinct future histopathological and immunohistochemical studies. For this study, some lymphoid (retropharyngeal lymph node, spleen) and nervous (frontal cortex, cerebellum, obex and medulla oblongata) samples were used. The postmortem interval between death and tissue processing was no less than 1 h. 

### 4.2. Experimental Design

First, for the selection of preclinical animals, the presence of pathological prion protein in biopsies was detected by immunohistochemistry. In a few cases, it was necessary to perform Protein Misfolding Cyclic Amplification (PMCA) and Dot Blot (DB) techniques on lymphoid tissues in order to confirm the in vivo diagnosis.

In order to assess the effect of DEX treatment on the animals included, neuropathological lesions were observed after Hematoxylin-eosin (H-E) staining; glial alterations (astrogliosis and microgliosis) were valued by immunohistochemical techniques for GFAP and IBA-1 detection, respectively; and variations in communication via cytokines released by glial cells were studied by means of selected cytokines detection by immunohistochemical techniques. 

### 4.3. Methodology

#### 4.3.1. Protein Misfolding Cyclic Amplification (PMCA)

Samples from lymphoid tissue (spleen or retropharyngeal lymph node) from 5 sheep whose status was doubtful after assessing pathological prion protein in RAMALT biopsies were used for this technique. Tg338 ovinized mice were used to prepare PMCA substrates. PMCA was performed as previously described [62,63]. Briefly, PMCA reactions (50 μL final volume) were seeded with 5 μL of each ovine sample. Then, they were subjected to 3 amplification rounds, each comprising 96 cycles (10 s sonication, 14 min and 50 s incubation at 39.5 °C) in a Qsonica700 device. After each round, reaction products (1 volume) were mixed with fresh substrate (9 volumes) to seed the following round. PMCA reaction products were analyzed by Dot Blot for the presence of pathological prion protein. 

#### 4.3.2. Dot Blot (DB)

Briefly, 18 µL of the product of PMCA from each sample was mixed with proteinase K (18 mg/mL, Roche, Reinach, Switzerland) and incubated for 1 h at 37 °C. Then, 25 µL Laemli (Bio-Rad, Hercules, CA, USA) was added to each sample. After several washes with 1% SDS, transference to a nitrocellulose membrane was carried out. The membrane was first blocked with 2% PBS-BSA before primary antibody anti-pathological prion protein was added (Sha 31, 1:10.000 for 30 min). After washing, the secondary antibody was added and the final evaluation was performed. 

#### 4.3.3. Histopathological (H-E) Studies

Hematoxylin-eosin (H-E) staining was applied on paraffin-embedded 4 μm sections in order to visualize the neuropathological lesions in different brain areas: the medulla oblongata (MO), the obex (O), the cerebellum (Cb) and the frontal cortex (Fc). As previously published by the group [14,19,20], neuronal vacuolation and spongiosis were assessed by counting the number of vacuoles present in the grey matter from each section; they were scored from 0 (minimum) to 4 (maximum) by two independent observers.

#### 4.3.4. Immunohistochemical Techniques (IHC)

Immunohistochemistry (IHC) was carried out in order to assess the accumulation of pathological prion protein, astroglial/microglial activation, and neuroinflammatory cytokines released by both glial populations. All IHC studies were developed on four brain areas (Fc, Cb, O and MO) from each untreated and treated animal. 

After specific pre-treatments for antigen retrieval, IHC protocols using specific primary antibodies against pathological prion protein and glial markers were applied. EnVision system (DAKO, Glostrup, Denmark) and diaminobenzidine (DAB; DAKO, Glostrup, Denmark) were used as the visualization system and chromogen, respectively. Hematoxylin counterstaining and mounting in DPX was performed on all sections. Table 3 summarizes each primary antibody and protocol used.

##### Pathological Prion Protein (PrP^sc^) Detection

As previously published [65], 98% formic acid immersion for 15 min, proteinase K (4 μg/mL; Roche, Reinach, Switzerland) treatment for 15 min at 37 °C, and hydrated heating for 20 min followed by endogenous peroxidase blocking (DAKO, Glostrup, Denmark) for 5 min and incubation with monoclonal antibody L42 (1:500, 30 min RT; DAKO, Glostrup, Denmark) were performed. 

##### Glial Fibrillary Acidic Protein (GFAP) Detection for Astrogliosis 

After endogenous peroxidase blocking (DAKO, Glostrup, Denmark) for 5 min, slides were incubated with a polyclonal antibody against glial fibrillary acidic protein (GFAP, 1:500, 30 min RT, DAKO, Glostrup, Denmark).

##### Ionized Calcium-Binding Adaptor Molecule-1 (IBA-1) Detection for Microgliosis 

Heat treatment for 20 min was necessary before endogenous peroxidase blocking (DAKO, Glostrup, Denmark) for 5 min. Afterwards, sections were incubated with a polyclonal antibody against ionized calcium binding adaptor molecule 1 (IBA-1, at 1:1.000; overnight 4 °C; WAKO, USA).

##### IL-1α, IL-1R, IL-6 and IFNγR Detection

A pre-treatment consisting of hydrated heating at 121 °C in 10% citrate buffer for 20 min preceded the endogenous peroxidase blocking (DAKO, Glostrup, Denmark) for 5 min and incubation with different primary antibodies: polyclonal IL-1α (1:100, overnight 4 °C; ThermoFisher Scientific, Waltham, MA, USA), polyclonal IL-1RN (1:100, overnight 4 °C; Sigma, St. Louis, MO, USA), monoclonal 8H12 (1:20, overnight 4 °C; ThermoFisher Scientific, Waltham, MA, USA), or polyclonal IFNGR1 (1:1200, overnight 4 °C; ThermoFisher Scientific, Waltham, MA, USA) were used.

##### IL-2R, IL-10R and TNFαR Detection

A pre-treatment consisting of hydrated heating at 96 °C in 10% citrate buffer for 20 min preceded the endogenous peroxidase blocking (DAKO, Glostrup, Denmark) for 5 min and incubation with different primary monoclonal antibodies: IL-2R.1 (1:1.000, overnight 4 °C; ThermoFisher Scientific, Waltham, MA, USA), OTI1D10 (1:250, overnight 4 °C; ThermoFisher Scientific, Waltham, MA, USA), or Ber-H2 (ready to use, 30 min RT; DAKO, Glostrup, Denmark) were used.

### 4.4. Statistical Analysis

For the results provided by IHC techniques, the normality of distribution was first tested with the Kolmogorov–Smirnov test. The non-parametric Mann–Whitney U test was performed to assess quantitative differences between treated and non-treated groups. 

SPSS software (SPSS Statistics for Windows, Version 17.0) was used for all analyses and significance in all cases was considered at * *p* ≤ 0.05, ** *p* ≤ 0.01 and *** *p* ≤ 0.001. All graphs were performed with GraphPad Prism 6.0. The data presented in Figures are expressed as means and the standard error of the mean (mean ± SEM).

## 5. Conclusions

The glucocorticoid treatment applied here directly influenced the neuroglial response in preclinical sheep naturally affected by scrapie. This suggests that the initial stage of the disease could constitute a therapeutic window that is not available in the clinical stage, in which animals present an impaired astroglial response and irreversible neurodegeneration. 

This neuroglia-mediated immune response confirms the occurrence of neuroinflammation in neurodegeneration, starting in the very early stages of scrapie infection in the field. It seems that this system is a defensive mechanism displayed by the whole encephalon in pathological conditions, producing an early attempt to re-establish homeostasis, but finally failing in the advanced stages of the disease.

## Figures and Tables

**Figure 1 ijms-21-05779-f001:**
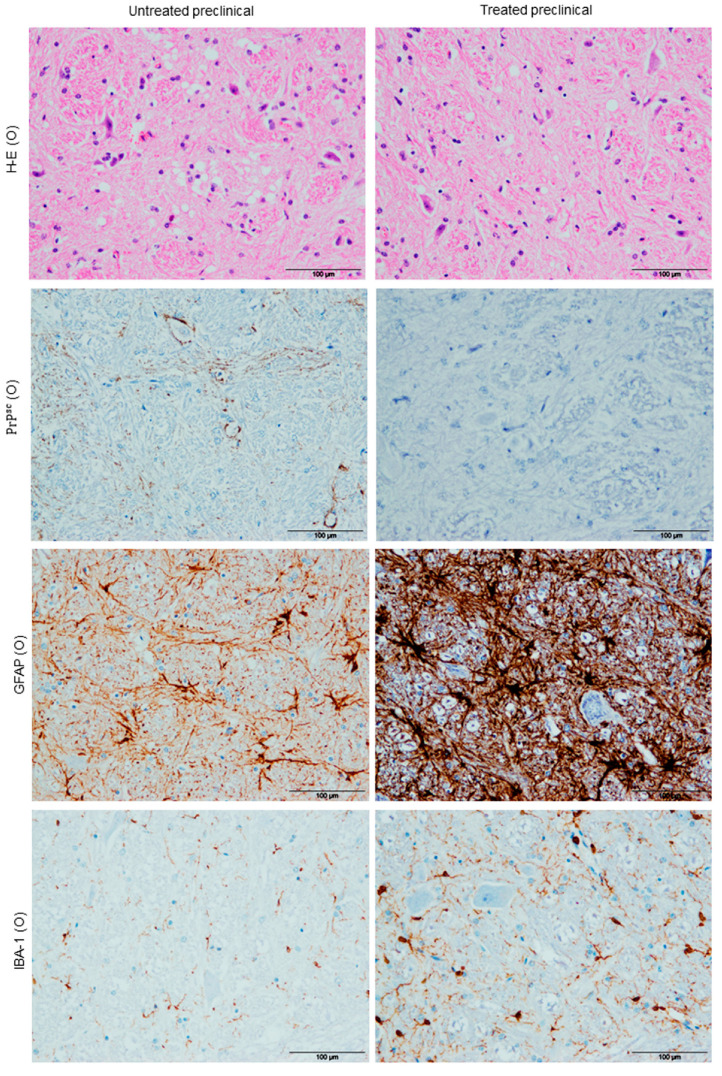
Neuropathological findings. Dexamethasone (DEX) treatment in preclinical sheep was associated with a reduction in vacuolation and pathological prion protein (PrP^sc^) deposits (as illustrated in obex O), while it was associated with a huge increase in astrogliosis and microgliosis in this same brain area. Scale bars: 100 µm.

**Figure 2 ijms-21-05779-f002:**
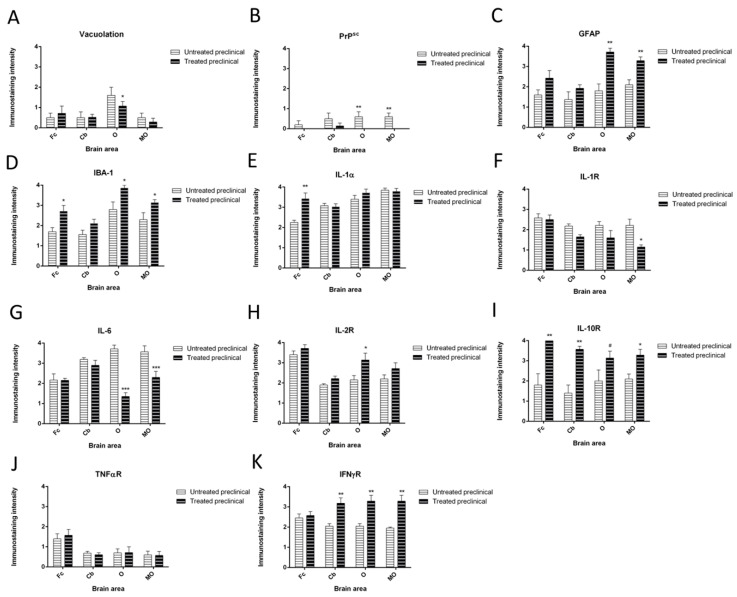
Statistical data concerning the variables assessed in the four brain areas (frontal cortex (Fc), cerebellum (Cb), obex (O), medulla oblongata (MO)) from DEX-treated and non-treated animals: (**A**) Vacuolation; (**B**) PrP^sc^ deposition; (**C**) Astrogliosis; (**D**) Microglial proliferation; (**E**) IL-1; (**F**) IL-1R; (**G**) IL-2R; (**H**) IL-6; (**I**) IL-10R; (**J**) TNFαR; (**K**) IFNγR.

**Figure 3 ijms-21-05779-f003:**
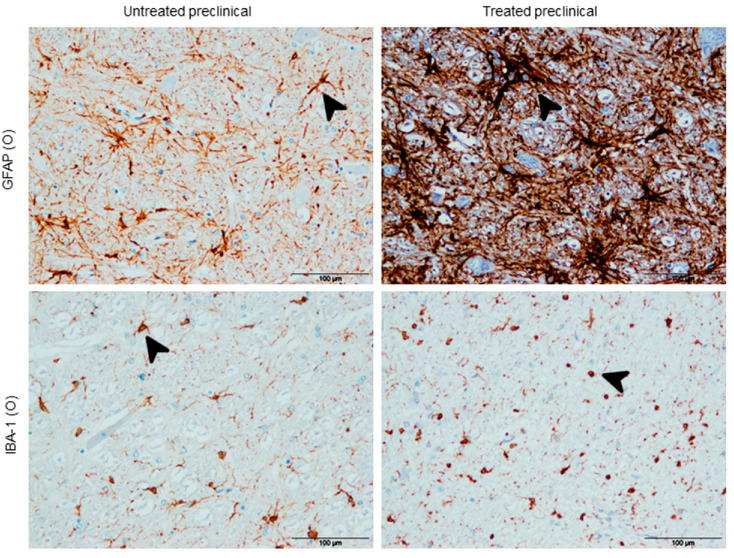
Morphological changes in neuroglia associated with treatment. Hypertrophic instead of stellate astroglial morphology (GFAP findings), and amoeboid instead of ramified microglial morphology (IBA-1 findings; see arrows). Scale bars: 100 µm. Obex (O).

**Figure 4 ijms-21-05779-f004:**
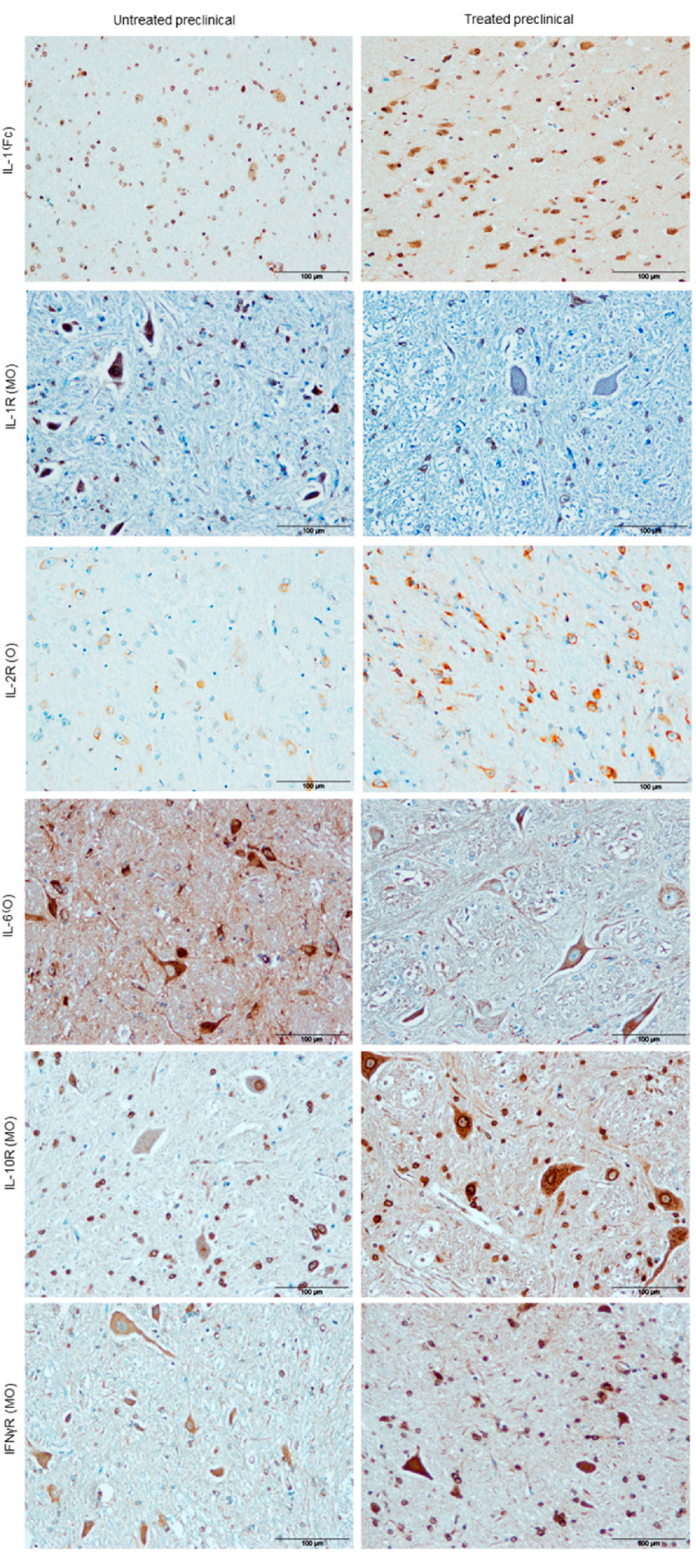
Illustration of cases (one area per case per marker) where differences in immunostaining for different cytokines (IL-1α, IL-1R, IL-2R, IL-6, IL-10R and IFNγR) associated with DEX treatment were evident. Scale bars: 100 µm. Frontal cortex (Fc), obex (O), medulla oblongata (MO).

**Figure 5 ijms-21-05779-f005:**
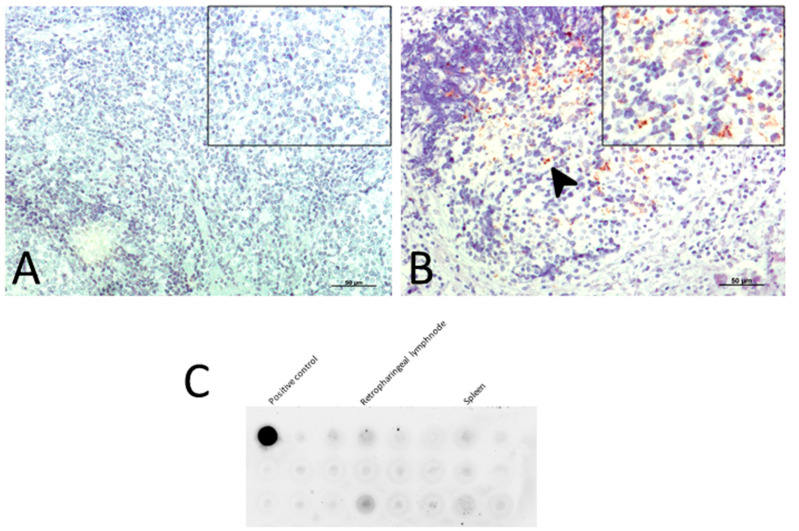
In vivo diagnosis: (**A**) using IHC for PrP^sc^ on biopsies of recto-anal mucosa associated lymphoid tissue (RAMALT) from healthy controls; and (**B**) from sheep presenting no clinical signs but PrP^sc^ deposits (arrow; preclinical stage of scrapie). Scale bars: 50 µm. (**C**) By Dot Blot on a positive retropharyngeal lymph node (product of amplification by Protein Misfolding Cyclic Amplification, PMCA).

**Table 1 ijms-21-05779-t001:** Summary of statistical results obtained in this study.

Marker Assessed	DEX Effect	Brain Area	Statistical Significance
Vacuolation	Decrease	O	* *p* ≤ 0.05
Prion protein deposition	Decrease	O MO	** *p* ≤ 0.01 ** *p* ≤ 0.01
GFAP	Increase	O MO	** *p* ≤ 0.01 ** *p* ≤ 0.01
IBA-1	Increase	Fc Cb O MO	* *p* ≤ 0.05 *# p* = 0.08 * *p* ≤ 0.05 * *p* ≤ 0.05
IL-1α	Increase	Fc	** *p* ≤ 0.01
IL-1R	Decrease	MO	* *p* ≤ 0.05
IL-2R	Increase	O	* *p* ≤ 0.05
IL-6	Decrease	O MO	*** *p* ≤ 0.001 *** *p* ≤ 0.001
IL-10R	Increase	Fc Cb O MO	** *p* ≤ 0.01 ** *p* ≤ 0.01 # *p* = 0.07 * *p* ≤ 0.05
TNFαR	No changes		*p* > 0.05
IFNγR	Increase	Cb O MO	** *p* ≤ 0.01 ** *p* ≤ 0.01 ** *p* ≤ 0.01

Frontal cortex (Fc), cerebellum (Cb), obex (O), medulla oblongata (MO).

**Table 2 ijms-21-05779-t002:** Summary of data corresponding to animal samples included in the study.

Sheep No.	PRNP Genotype	Age (Years)	Group	Treatment and Duration
1	ARQ/ARQ	3	Preclinical	Untreated
2	ARQ/ARQ	3	Preclinical	Untreated
3	ARQ/ARQ	6	Preclinical	Untreated
4	ARQ/ARQ	3	Preclinical	Untreated
5	ARK/ARQ	4	Preclinical	Untreated
6	ARQ/ARQ	8	Preclinical	Treated (7 months)
7	ARQ/ARQ	3	Preclinical	Treated (11 months)
8	ARQ/ARQ	7	Preclinical	Treated (12 months)
9	ARQ/ARQ	8	Preclinical	Treated (13 months)
10	ARQ/ARQ	5	Preclinical	Treated (13 months)
11	ARQ/ARQ	5	Preclinical	Treated (14 months)
12	ARQ/ARQ	6	Preclinical	Treated (17 months)

**Table 3 ijms-21-05779-t003:** Primary antibodies used for different immunohistochemical techniques and the retrieval methods applied for each of them.

Antibody	Antigen	Type	Dilution	Retrieval Method	Source
L42	Prion protein	Monoclonal	1:500	Formic acid 15 min Proteinase K 15 min Heat treatment 20 min Peroxidase blocking	DAKO
Anti- GFAP	GFAP	Polyclonal	1:500	Peroxidase blocking	DAKO
Anti-IBA-1	IBA-1	Polyclonal	1:1.000	Heat treatment 20 min Peroxidase blocking	WAKO
IL-1 alpha	IL-1	Polyclonal	1:100	Autoclave 121 °C (citrate buffer 10%)	ThermoFisher
Anti-IL-1RN	IL-1R	Polyclonal	1:100	Autoclave 121 °C (citrate buffer 10%)	Sigma
IL-2R.1	IL-2R	Monoclonal	1:1.000	PTLink 96 °C	ThermoFisher
8H12	IL-6	Monoclonal	1:20	Autoclave 121 °C (citrate buffer 10%)	ThermoFisher
OTI1D10	IL-10R	Monoclonal	1:250	PTLink 96 °C	ThermoFisher
Ber-H2	TNFαR	Monoclonal	Ready to use	PTLink 96 °C	DAKO
IFNGR1	IFNγR	Polyclonal	1:200	Autoclave 121 °C (citrate buffer 10%)	ThermoFisher

Immunostained sections were scored by two independent operators on a scale ranging from 0 (minimum) to 4 (maximum), as previously described [14,19,20], regarding the intensity of pathological prion protein accumulation, and astroglial and microglial activation. Morphological glial alterations were also evaluated in 10 microscopic fields in each brain region. For neuroinflammatory markers, observers scored the intensity of immunostaining by counting positive cells [64] in 5 microscopic fields in each brain region examined.

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
