# Peer review of "Neuroimmune Response in Natural Preclinical Scrapie after Dexamethasone Treatment"

_ijms, 2020, doi:10.3390/ijms21165779_

Round 1

Reviewer 1 Report

In the present MS, Guijarro et al, presented the neuroinmune response obtained after dexamethasone treatment in a preclinical model of scrapie in sheep. In this study, authors revealed that dexamethasone exert an effect on the glial activation or glial communication, among others findings. The positive effect of dexamethasone in neurominmmune response is an interesting finding, but some important questions should be elucidated.

Authors revealed a hypertrophic glial morphology in DEX-treated sheep compared to untreated sheep in all brain areas. This idea is explained in the same manner for the morphology of microglia. However, morphological deficit of astrocytes is a classical sign in the early onset of many neurodegeneration illness (Alzheimer, Huntington's Disease, etc).  It would be possible that the onset of the scrapie was different for untreated and treated groups and this effect was a natural process in a temporal window in the development of scrapie?. What clinical signs were considered for the onset of this pathology? Have the authors results from control groups without scrapie illness?

The authors have included an animal treated for 7 months that differ significantly of the rest of treated groups (11,12, 13 and until 17 months), did they find some important (clinical or histological) differences in the results in the group treated at 7 months?

The findings about “hypertrophic glia” are very attractive and they should have quantified in the present study.

Regarding “material and methods”, authors should to include a section of “experimental design” were they detail the aim of each experiment. For the rest of sections, authors should include a title more descriptive as “ Histopathological studies”

Please, in the results, each section should include a titled related with the finding found.

All the text should be justified (homogenously between the margins).

Other points to consider:

  • The images relative to the diagnosis present low quality and they seem to be captured in different objectives or zoom. Please, include a representative image with a zoom in the superior right quadrant.
  • Images from “dot blot technique” should be included in the MS
  • In the legend of each table or figure, abbreviation should be including.
  • The more relevant figure of the MS are really small (figure 2). Please, increase the size of each graph. In addition, lthe letter should be in correct order, from the left to the righ (first line: A, B, C; second line: D, E, F….etc)

Author Response

In the present MS, Guijarro et al, presented the neuroinmune response obtained after dexamethasone treatment in a preclinical model of scrapie in sheep. In this study, authors revealed that dexamethasone exert an effect on the glial activation or glial communication, among others findings. The positive effect of dexamethasone in neurominmmune response is an interesting finding, but some important questions should be elucidated.

We thank the Reviewer for his/her positive comments about the interest of the paper and helpful suggestions that the Authors address below.

Authors revealed a hypertrophic glial morphology in DEX-treated sheep compared to untreated sheep in all brain areas. This idea is explained in the same manner for the morphology of microglia. However, morphological deficit of astrocytes is a classical sign in the early onset of many neurodegeneration illness (Alzheimer, Huntington's Disease, etc).  It would be possible that the onset of the scrapie was different for untreated and treated groups and this effect was a natural process in a temporal window in the development of scrapie?. What clinical signs were considered for the onset of this pathology? Have the authors results from control groups without scrapie illness?

Reviewer’s comment is really interesting. Scrapie affected animals at preclinical stage that had been not treated were used as controls precisely because of the Authors wanted to be sure of that changes were due to treatment and not to the onset of disease. The Authors noticed that this idea was not clear and it has been clarified.

A total of 12 female Rasa Aragonesa crosses Scrapie pre-clinically affected were included in the present study. Two different groups were studied: non-treated (n = 5) and DEX-treated (n = 7) animals.

Taking into account the Reviewer’s question, clinical signs considered for the onset of this pathology have been included in the text. 

These animals did no exhibit any clinical signs of scrapie before inclusion in the study and throughout the experiment. An exhaustive clinical diagnosis based on classical signs such as pruritus, tremor, locomotor incoordination and behavioural changes was carried out on all of them.

As the Reviewer suggests, the Authors tried to add a control group without Scrapie illness, but as it is really hard to get access to preclinical animals, it was no possible to finally include this control group. 

The authors have included an animal treated for 7 months that differ significantly of the rest of treated groups (11, 12, 13 and until 17 months), did they find some important (clinical or histological) differences in the results in the group treated at 7 months?

The Authors considers this observation very justified, but no differences were found between this and the rest of animals. In fact, in case it would have existed, it would be appreciable in bars (SEM) represented in Figure 3.  

The findings about “hypertrophic glia” are very attractive and they should have quantified in the present study.

Following Reviewer’s suggestion, comment about quantification of hypertrophic glia has been added in the text. It was observed in the 10 microscopic fields assessed in each brain region in 100% of dexamethasone treated animals (7 sheep).

Morphologically, GFAP immunolabeling showed a hypertrophic morphology in 100% of treated sheep compared to astrocytes in untreated animals in all brain regions, which appeared to be more stellate (Fig 3).

Regarding “material and methods”, authors should to include a section of “experimental design” were they detail the aim of each experiment. For the rest of sections, authors should include a title more descriptive as “Histopathological studies”

In agreement with Reviewer’s recommendation, a section of ‘experimental design’ has been included in Material and Methods section to detail the aim of each experiment.

Experimental design

First, for selection of preclinical animals, the presence of pathological prion protein in biopsies was detected by immunohistochemistry. In a few cases, it was necessary to perform Protein Misfolding Cyclic Amplification (PMCA) and Dot Blot (DB) techniques on lymphoid tissues in order to confirm the in vivo diagnosis.

In order to assess the effect of DEX treatment in animals included, neuropathological lesions were observed after Hematoxylin-eosin (H-E) staining; glial alterations (astrogliosis and microgliosis) were valued by immunohistochemical techniques for GFAP and IBA-1 detection, respectively; and variations in communication via cytokines released by glial cells were studied by means of selected cytokines detection by immunohistochemical techniques.

Moreover, in the rest of sections the title has been substituted by another one more descriptive.

Please, in the results, each section should include a titled related with the finding found.

According with Reviewer’s suggestion, a title has been included for each section.

All the text should be justified (homogenously between the margins).

The text has been justified as Reviewer suggested.

Other points to consider: 

  • The images relative to the diagnosis present low quality and they seem to be captured in different objectives or zoom. Please, include a representative image with a zoom in the superior right quadrant.

The Figures have been replaced by those suggested by Reviewer (representative area + zoom).

  • Images from “dot blot technique” should be included in the MS

Dot blot technique has been included in Fig 5C in the MS.

  • In the legend of each table or figure, abbreviation should be including.

Abbreviations have been included in each Table and Figure.

  • The more relevant figure of the MS are really small (figure 2). Please, increase the size of each graph. In addition, the letter should be in correct order, from the left to the righ (first line: A, B, C; second line: D, E, F….etc)

Following the Reviewer’s suggestion, the size of each graph has been increased in Fig 2 and the letters have been reordered.

Reviewer 2 Report

The manuscript “Neuroimmune response in natural preclinical scrapie after dexamethasone treatment” highlights the critical role of glucocorticoid treatment affecting immune response. Here are the few points that should be addressed to improve the overall quality of the manuscript

Please number the heading and subheadings throughout the manuscript

Please include data regarding the initial assessment for establishing the disease model at the beginning, clubbing table and figures together.  

It would be appreciative if authors could include a diagrammatic representation of the experimental paradigm explaining about the groups, route of treatment, duration of treatment in a precise manner for better understanding

Please provide a table giving a description of the number of samples collected (~80 as mentioned in the manuscript).

Immunohistochemistry – Clearly indicate in text how many regions per sections were analyzed for each marker including statistical significance in the figure legends

IBA-1 is a general marker for the macrophages, staining resident microglia as well as monocyte-derived macrophages. There is an increase in astrocytic activity (also involved in maintaining the BBB) following DEX treatment. Hence it would be important to distinguish whether it was brain-resident microglia or peripheral monocytes (macrophages). TMEM119 is a marker that could solve this issue. In addition, activation of microglia could be verified by CD68 as well

Author Response

The manuscript “Neuroimmune response in natural preclinical scrapie after dexamethasone treatment” highlights the critical role of glucocorticoid treatment affecting immune response. Here are the few points that should be addressed to improve the overall quality of the manuscript.

We thank the Reviewer for his/her positive comments about the interest of the paper and helpful suggestions that the Authors address below.

Please number the heading and subheadings throughout the manuscript.

Following the Reviewer’s suggestion, heading and subheadings have been numbered.

Please include data regarding the initial assessment for establishing the disease model at the beginning, clubbing table and figures together.  

Taking into account the Reviewer’s question, initial clinical assessment for disease model has been included in the text. 

These animals did no exhibit any clinical signs of scrapie before inclusion in the study and throughout the experiment. An exhaustive clinical diagnosis based on classical signs such as pruritus, tremor, locomotor incoordination and behavioral changes was carried out on all of them.

It would be appreciative if authors could include a diagrammatic representation of the experimental paradigm explaining about the groups, route of treatment, duration of treatment in a precise manner for better understanding.

According with Reviewer’s request, the experimental design has been described in Material and Methods section. 

4.2 Experimental design

First, for selection of preclinical animals, the presence of pathological prion protein in biopsies was detected by immunohistochemistry. In a few cases, it was necessary to perform Protein Misfolding Cyclic Amplification (PMCA) and Dot Blot (DB) techniques on lymphoid tissues in order to confirm the in vivo diagnosis.

In order to assess the effect of DEX treatment in animals included, neuropathological lesions were observed after Hematoxylin-eosin (H-E) staining; glial alterations (astrogliosis and microgliosis) were valued by immunohistochemical techniques for GFAP and IBA-1 detection, respectively; and variations in communication via cytokines released by glial cells were studied by means of selected cytokines detection by immunohistochemical techniques.

In case Reviewer considers more appropriate to add a diagrammatic representation, please, do not hesitate in requiring it. 

Please provide a table giving a description of the number of samples collected (~80 as mentioned in the manuscript).

In accordance with Reviewer’s request, the number of samples collected has been indicated in text.

A total of 80 samples were collected (including different tissues: digestive, lymphoid, nervous and other viscera) and each one was fixed by immersion in 4% paraformaldehyde for this and other distinct future histopathological and immunohistochemical studies. For this study, some lymphoid (retropharyngeal lymphonode, spleen) and nervous (frontal cortex, cerebellum, obex and medulla oblongata) samples were used.

Immunohistochemistry – Clearly indicate in text how many regions per sections were analyzed for each marker including statistical significance in the figure legends.

Following Reviewer’s suggestion, the number of regions analysed per section has been specified.

All IHC studies were developed on four brain areas (Fc, Cb, O, and MO) from each untreated and treated animal.

IBA-1 is a general marker for the macrophages, staining resident microglia as well as monocyte-derived macrophages. There is an increase in astrocytic activity (also involved in maintaining the BBB) following DEX treatment. Hence it would be important to distinguish whether it was brain-resident microglia or peripheral monocytes (macrophages). TMEM119 is a marker that could solve this issue. In addition, activation of microglia could be verified by CD68 as well

The Authors are in total agreement with Reviewer’s comment. In fact, CD68 marker has been applied in some of our previous studies to assess activated microglia. However, in this occasion, the Authors decided to use IBA-1 in order to assess not only quantitative but also morphological variations since in our opinion is a very important aspect to take into account in neurodegenerative process. 

Round 2

Reviewer 2 Report

The revised version of the manuscript has addressed all the aforementioned points including that for IBA1 staining of microglia. 

With all the comments addressed, the manuscript seems fit for publication in its present form.